# MiXR-Interact: Mixed Reality Interaction Dataset for Gaze, Hand, and Body

N. B. Takele[1,2], D. Delehelle[1,2], Y. Kim[1], Y. T. Tefera[1], N. Deshpande[3],
D. G. Caldwell[1], J. Ortiz[1], and C. Recchiuto[2]

[1]Istituto Italiano di Tecnologia (IIT), Via Morego 30, 16163 Genova, Italy
[2]University of Genova, via all'Opera Pia 13, 16145, Genova, Italy
[3] School of Computer Science, University of Nottingham, Nottingham, NG8 1BB, UK

*Abstract*—This paper presents MiXR-Interact, a dataset providing motion tracking data for users' interactions in mixed reality (MR) environments, focusing on tracking their gaze, upper body movements, and hand gestures. The dataset is based on the Meta Quest Pro headset, offering an easy-to-use resource for researchers and developers working in MR and human-computer interaction (HCI). MiXR-Interact focuses on collecting natural and precise interactions with virtual objects, with three core interaction types: pushing, pointing, and grasping. To ensure robustness and generalization, each interaction is performed across six distinct directions, reflecting a diverse range of movement trajectories relative to the user's body. This directional diversity provides critical insights into how users approach and engage with virtual objects from multiple angles. In addition, to precisely track contact points during interactions, 17 key contact points are defined for each direction and are labeled. These contact points are used as reference markers to accurately localize and quantify the joint-to-object contact points for each interaction type and direction. In addition to providing the dataset, this paper evaluates the quality and precision of the collected dataset in MR through a set of evaluation metrics. These metrics assess critical aspects of interaction performance, including Trajectory Similarity, Joint Orientation, and Joint-to-Contact Alignment. It also details the theoretical and implementation considerations for dataset collection, offering valuable insights for applications in MR and human-robot interaction (HRI).

*Index Terms*—Mixed-Reality, Body tracking, Human-robot interaction

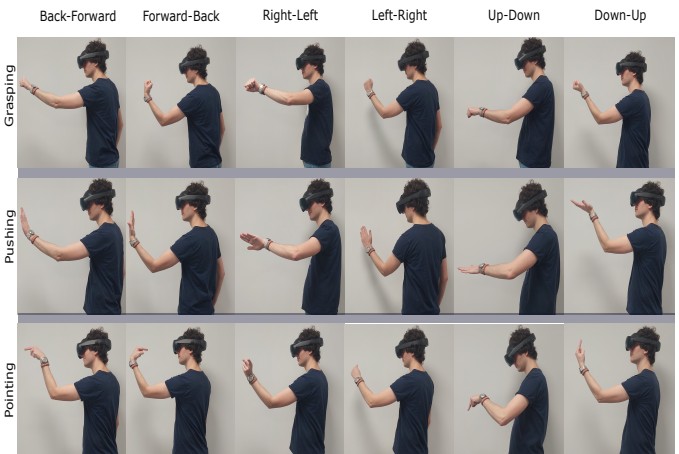

Fig. 1: MiXR-Interact: Data Collection Setup for Capturing Gaze, Upper Body Movements, and Hand Interactions in a Mixed Reality Environment

## I. INTRODUCTION

Gaze direction, hand gestures, and full-body positions are part of the most important information cues for natural human communication and interactions. They allow us to perceive and interpret information even before it is verbally conveyed. These interaction methods are particularly critical in applications like Mixed Reality (MR), where creating an immersive experience demands a seamless and intuitive flow of information [1]. Complex MR applications, such as immersive telepresence and teleoperation, rely on exchanging rich and detailed interaction data to maintain a high level of realism and engagement across networked environments. Telepresence enables users to experience a distant environment as if they were physically present, while teleoperation involves the remote control of machines or systems, often in scenarios where direct human involvement is impractical or unsafe [2].

This applications faces challenges like latency, information representation, and limited contextual awareness due to the physical distance between the user and the remote systems, which can disrupt immersion and control. Despite extensive research in these applications, the physical separation remains a significant obstacle to achieve high-level seamless presence and interaction across remote and virtual environments [3]–[5].

To address these challenges, state predictive mechanisms could be used to update the MR elements. User's body state such as facial expressions, gestures, posture, and mannerisms can be used to anticipate and communicate. This states are powerful form of nonverbal communication cues that often occurs instinctively, without deliberate thought [6]. During interactions or conversations, people instinctively observe and interpret subtle movements, gestures, and expressions from those around them. For example, a slight adjustment in posture, a raised eyebrow, or crossed arms can convey a range of emotions such as confidence, curiosity, or defensiveness. By tuning into these signals, people naturally form impressions, predict behaviors, and anticipate what others might say or do next. A fascinating example of this phenomenon occurs in musical ensembles, where musicians use nonverbal communication, particularly eye gaze, to convey their intentions and interpret co-performers [7]. Techniques such as machine learning models could be trained on body states and could be

used to anticipate user intent.

Meta Quest Pro was chosen for its affordability and widespread use in MR research, making it a practical foundation for dataset development. Furthermore, the methods and protocols used to create the dataset are hardware-agnostic, allowing them to adapt seamlessly to other MR devices with similar tracking capabilities. This ensures broader applicability across diverse research environments.

The integration of gaze, upper body movements, and hand interactions is essential for advancing MR, HCI, and HRI. Each feature provides unique insights: gaze direction reveals the user's focus, upper body movements indicate posture and spatial orientation, and hand interactions capture fine motor skills required for object manipulation. Studying these elements separately often misses the broader context of user behavior. By combining them, it becomes possible to gain a deeper understanding of user interactions and to design systems that respond naturally and intuitively.

This paper presents MiXR-Interact, a novel dataset that combines gaze, upper body movements, hand movements, and virtual object poses to improve the composition of existing datasets and provide researchers with a rigorously evaluated resource for advancing MR, HCI, and HRI development. The dataset has been made openly available to enable broader access and support further research in these fields [1]. Additionally, the paper explores various methodologies and considerations for collecting datasets in immersive environments, focusing on the complexities and nuances of capturing user interactions.

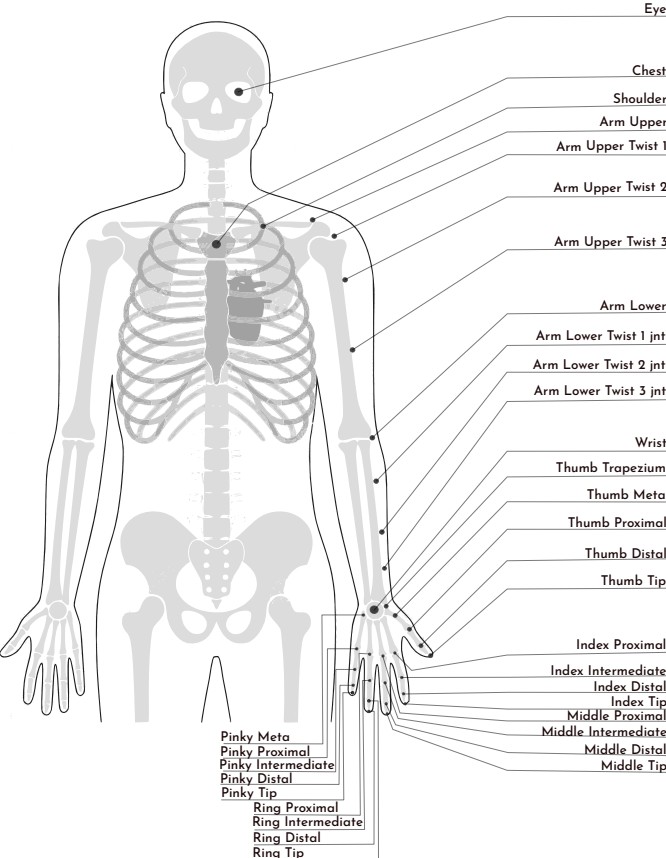

Fig. 2: Human Body Pose and Joints

## II. RELATED WORK

Several researchers showed that during interaction, humans often begin looking towards an object before initiating movement [8]. This suggests that gaze features are valuable indicators for intent prediction and early human activity detection, playing a key role in visual information processing and measuring attention, interest, and arousal. For instance, a study introduced in [9], [10], presented a dataset of joint poses and gazes to enhance prediction accuracy. It used gaze and motion features to estimate user intentions and predict current actions and target objects. Furthermore, the work described in [11] has been aimed to develop eye-hand coordination and gaze-based intention recognition to improve performance in teleoperated pick-and-place tasks. By tracking a user's hand, a system was developed to predict both the timing and likelihood of specific virtual objects colliding with the user [12].

Various sensors have been used to track human body movements and gaze. Electromyography (EMG) and inertial measurement units (IMUs) are commonly used to capture human body poses [13]. RGB-D cameras have been utilized to monitor hand movements [14], while the Microsoft Kinect Sensor has been used to track joint movements related to nonverbal cues [15]. For gaze tracking, researchers have developed head-mounted devices combined with a small endoscopic

camera, infrared light, and a mobile phone to provide cost-effective and accurate vision tracking solutions and estimate the gaze point [16]. In the context of MR, predicting hand-object interaction involves using a recorded history of hand points and virtual object poses [17].

Despite substantial advancements in MR, HCI, or HRI, a significant gap persists in datasets that integrate tracking data for gaze, upper body motion, hand movements, and virtual object poses within MR environments. To bridge this gap, we introduce MiXR-Interact to enhance the composition of existing datasets and provide researchers with a rigorously evaluated resource.

## III. BUILDING THE DATASET

When collecting the dataset, it was important to establish clear objectives to ensure a systematic and purposeful approach to data acquisition. The primary objective defined for this dataset was to enable the identification of the most likely contact points where a user intends to interact and to allow for the rapid localization of these points. This objective makes sure that the dataset supports systems in accurately predicting user intent in real-time scenarios. The secondary goal was to provide insights into the type of interaction the user is likely to perform. This goes beyond merely identifying contact locations to include an understanding of the nature of the interaction itself. For instance, the dataset should facilitate

[1] github.io/MiXR-Interact-Dataset

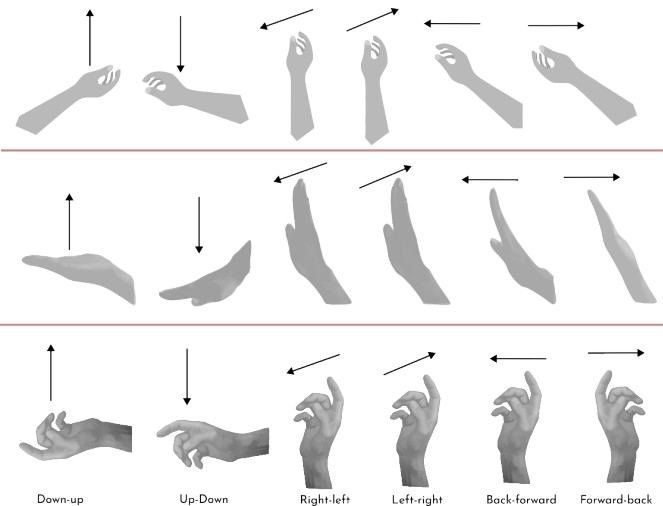

Fig. 3: The six directional grasping, pushing, and pointing gestures

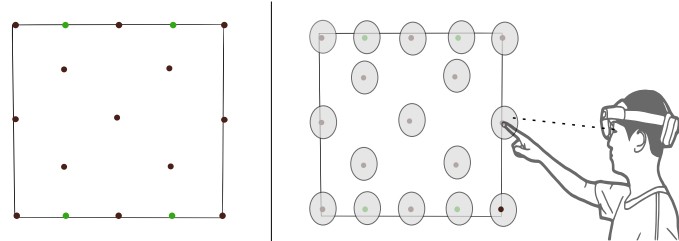

Fig. 4: Data collection setup: The left panel shows the contact points, while the right panel illustrates the pointing interaction.

the differentiation between actions such as a push, a grasp, or a point.

To achieve these goals, it is important to understand the complexities of human anatomy and behavior. A thorough understanding of the human body, particularly the anatomy and dynamics of the eyes, upper body, and hand joints, is necessary. These elements play a central role in interactions, as the gaze directs attention, the upper body conveys movement intent, and the hand joints execute precise actions. The following sections provide an in-depth exploration of the relevant anatomical features and the characteristics of the dataset, highlighting how these aspects were defined and incorporated to create a robust resource for understanding and modeling human interactions.

### A. Anatomy of the Eye and Upper Body Joints in Interaction

We believe that tracking eye movements and upper body joints (Fig. 2) could play a crucial role in accurately interpreting and predicting human intentions. The human eye plays a critical role in perception and interaction through gaze behavior, signaling intent, attention, and emotional states. Its movements, controlled by six extraocular muscles, enable tracking and focus on objects, with visual information processed by photoreceptor cells in the retina, making gaze a key component of nonverbal communication [18], [19]. In addition, the upper body joints, particularly in the shoulders, elbows, wrists, hands, and fingers, are vital for recognizing human intent. Larger movements originate at the shoulder and elbow joints, while finer adjustments occur in the Metacarpophalangeal (MCP), Proximal Interphalangeal (PIP), and Distal Interphalangeal (DIP) joints, with the thumb's unique Carpometacarpal (CMC) joint enabling precision manipulation [20]. To capture this intent-driven motion, we collected the user's gaze position, along with the position and orientation of their body and hands as they interacted with virtual objects.

In addition to user data, detailed information about the MR environment is collected, including the 3D positions and orientations of all objects within the virtual space. It includes predefined contact poses, specific spatial locations, and configurations where user interactions, such as grasping, pointing, or pushing, are intended to occur.

### B. Dataset Characteristics

Maintaining consistency during data collection is crucial for ensuring that the resulting data is reliable, reproducible, and accurate. Inconsistencies in the data can introduce bias, reduce validity, and lead to unreliable conclusions. To achieve these goals, we have established three key characteristics that define the quality and integrity of the dataset:

1) **Diversity in interaction:** To ensure that this dataset captures a representative range of interactions, inspired by authors in [21], we defined three key interaction types: grasping, pushing, and pointing (Fig. 1). These interactions were selected because they encompass a broad spectrum of common human-object and human-environment engagements. In addition, these three interactions were performed in six distinct directions to capture a full range of motion: forward, backward, left-to-right, right-to-left, upward, and downward, as seen in Fig. 3.

2) **Consistency in collection:** A careful experimental protocol was defined to ensure that the collected data is consistent in flow and direction.

3) **Accuracy in contact point interaction:** To ensure accurate and precise contact point interactions during data collection, the contact points in MR must be large enough to facilitate easy interaction while remaining small enough to require precise contact.

## IV. DATA COLLECTION

### A. Data Collection Setup

This dataset and the collection mechanism are based on the Meta Quest Pro MR headset system. The headset supports gaze, hand, and body tracking, as well as motion controllers for interaction. During the collection, it was connected to a high-performance computer with an NVIDIA RTX 3090 GPU and an Intel i9 processor to handle the rendering of the MR environment and save the tracked data.

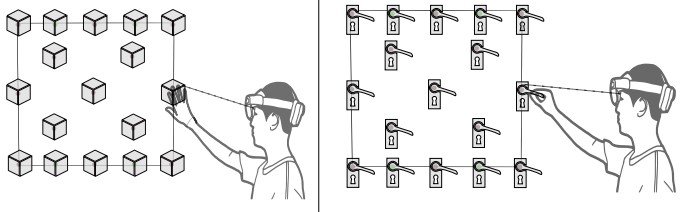

Fig. 5: Data collection setup: The left image illustrates the pushing interaction, while the right panel depicts the grasping.

The MR interface was created using the Unreal Engine 5. As illustrated in Fig. 4 (Left), participants interact with a plane that contains designated contact points. These contact points serve as anchors for the placement of various 3D mesh objects, each selected to the specific task at hand. The type and shape of the 3D mesh objects are selected to facilitate natural interaction, with common examples including spherical, rectangular, and handle-shaped meshes.

For instance, in the Pointing task, a spherical mesh is placed at the contact point, as shown in Fig. 4 (Right). The spherical shape is chosen to provide a clear and intuitive target for participants, allowing them to interact with the virtual plane through precise pointing gestures. For tasks that involve pushing, a rectangular mesh is positioned at the contact points, as depicted in Fig. 5 (Left). The larger surface area of the rectangular mesh enables participants to apply force with a broader contact region, supporting a more natural pushing action. In grasping tasks, a handle-shaped mesh, resembling a door handle, is placed at the contact points, as illustrated in Fig. 5 (Right). The handle-shaped design facilitates grasping actions, encouraging participants to employ natural hand poses and grip strategies similar to how they would interact with a real-world handle.

### B. Dataset Description

The "Eye Tracking API" enables developers to access gaze direction information captured by the Quest Pro's inward-facing cameras. This functionality provides insights into where a user is looking. Similarly, the "Body Tracking API" offers an upper-body skeleton model that leverages the positional data from the head and hands. However, it is important to note that the body tracking system is limited to the upper body and does not include data for the legs or lower body.

In Fig. 2, the tracked points for hand movements, upper body articulation, and eye gaze are illustrated, highlighting the specific regions covered by the APIs. For example, each hand has 23 distinct joints spanning the fingers and wrist. These joints are individually tracked to provide detailed data on both position and orientation. For gaze tracking, only the position data of the gaze origin and fixation point is collected, allowing the gaze direction to be easily calculated from these two points. Since this tracking information is generated as a time series dataset, it can be easily read, analyzed, and processed using a Pandas DataFrame.

### C. Data Collection Procedure

A total of 20 human subjects (5 females and 15 males) aged between 22 and 33 years participated in the study. The mean age of the sample population was $\mu = 26.35$ years, with a standard deviation of $\sigma = 2.71$ years. All subjects had a 20/20 or corrected vision, and the eye tracker was calibrated for all subjects. Based on the ITU-T [22] recommendation, subjects were made familiar with the setup, the MR headset, and the MR environment. Each subject performed all three interaction types, engaging with all 17 contact points in six distinct directions. To ensure consistency in the collected pose data, subjects were instructed to return their hand to a reset position before moving on to the next task. This step was critical for maintaining uniform trajectories across subjects. The experimenter controlled when the next contact point appeared, ensuring participants followed the correct sequence and pace to prevent rushing or skipping steps, maintaining the integrity of the dataset. All participants in this study were volunteers, and each provided informed consent by signing a consent form permitting the collection and use of their data for the specified research purposes. The data collection process was approved by the *Ethics Committee of Liguria Region*, under the protocol *IIT_ADVR_TELE01*.

## V. EVALUATION METRICS

In addition to the dataset, the paper introduces evaluation metrics for assessing trajectory similarity, joint orientation, and joint-to-contact position differences.

### A. Trajectory Similarity

When collecting data, joint trajectories are assumed to be similar for interactions occurring in a single, consistent direction. However, due to variations in movement speed, timing, and subtle deviations in spatial paths, direct point-by-point comparisons may not accurately capture the actual similarity between trajectories. To address these challenges, we used 3D Dynamic Time Warping (DTW) [23] as the primary evaluation metric to measure and quantify the similarity between 3D trajectories.

Unlike traditional Euclidean distance, which assumes fixed point-to-point correspondence at equal time intervals, 3D DTW provides a more flexible approach. It allows for non-linear alignment of the time axis by stretching, compressing, or shifting time to achieve the optimal alignment between two trajectories. This process ensures that trajectories that are similar but misaligned in time (e.g., one subject moving faster than another) can still be recognized as similar. The alignment is achieved by calculating the cumulative Euclidean distance between corresponding 3D points $[x, y, z]$ at each time step along the trajectories while simultaneously determining the best alignment path that minimizes the total distance. The final output of 3D DTW is a distance score, which serves as an indicator of similarity. A smaller DTW distance indicates that the trajectories are more similar, while larger distances reflect greater differences in spatial paths, timing, or both. For perfectly identical trajectories, the DTW distance is zero.

## B. Joint Orientation

Since the orientation of finger joints provides important insights into the direction of interaction, analyzing the joint orientations becomes important. Finger joints naturally align with the intended direction of motion or interaction, making their orientation a key indicator of user intent.

To facilitate this analysis, we first converted the quaternion representations of joint rotations into Euler angles, which include roll, pitch, and yaw orientations. This conversion allowed us to represent the joint orientations in a format that is more intuitive and easier to analyze geometrically. Next, we used the Rayleigh Hypothesis Test to evaluate the statistical distribution of these angles. This test is particularly well-suited for analyzing directional data, as it determines whether the observed angles exhibit a uniform distribution or show significant clustering around a preferred direction.

- **Null Hypothesis (H_0)**: The angles are uniformly distributed around the circle, meaning no preferred direction exists.
- **Alternative Hypothesis (H_a)**: The angles are not uniformly distributed, indicating the presence of a dominant or preferred direction.

If the null hypothesis is rejected, it suggests that the angles are not randomly distributed but are instead clustered around a specific angle, revealing a preferred orientation in the movement or interaction direction. By applying this method, we aimed to uncover patterns and tendencies in finger joint orientations that could be linked to specific types of interactions or movement intents.

## C. Joint to Contact Position Distance

Subjects interacted with a virtual object, where precise alignment between the subject's final joint position and the object's contact point was expected. Ideally, at the moment of interaction, the joint position should coincide perfectly with the designated contact point on the virtual object, ensuring no spatial gap between them. Achieving this alignment is crucial for accurate tracking, naturalistic interaction, and effective coordination with the virtual environment. To quantify this alignment, we used the Euclidean distance as the evaluation metric. Euclidean distance measures the straight-line distance between two points in 3D space. In this context, the two critical points are the final joint position and the corresponding contact point on the object.

As mentioned in the data collection section, a cubic mesh is used for pushing, while a spherical mesh is used for pointing, and a door handle model is used for grasping, all using simplified collision models to enhance the responsiveness of virtual objects. These simplified models are expected to introduce small deviations, and this evaluation aims to identify and analyze these discrepancies.

## VI. EVALUATION AND ANALYSIS

Figure 6 presents a frame-by-frame illustration of the grasping, pushing, and pointing interactions. The figure shows how

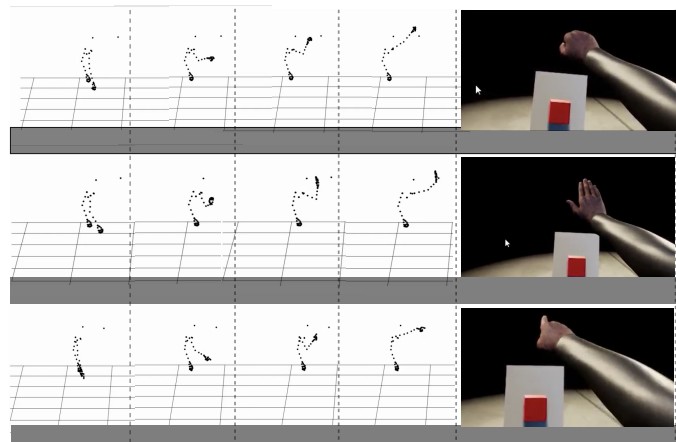

Fig. 6: Frame-by-frame illustration of grasping, pushing, and pointing interactions and their corresponding MR environment.

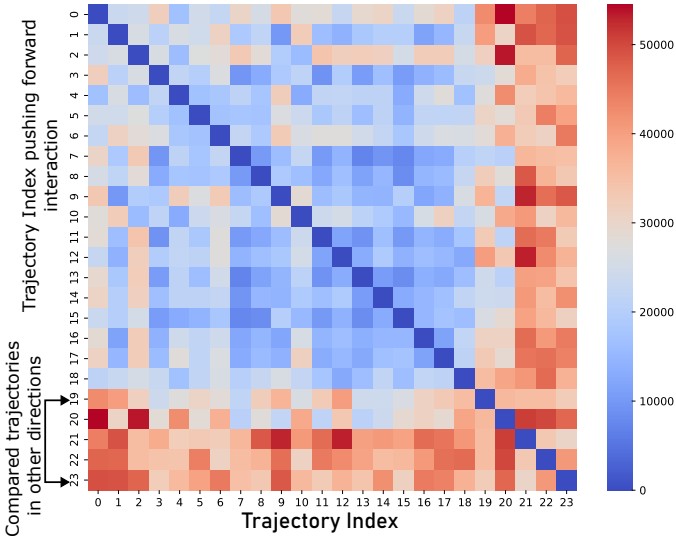

Fig. 7: Analysis of trajectory similarity using dynamic time warping for pushing interaction.

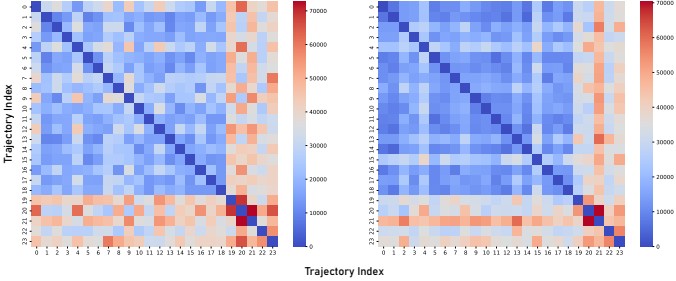

Fig. 8: Analysis of trajectory similarity using dynamic time warping for grasping in the left and pointing on the right interaction.

the joints are tracked throughout each interaction, demonstrating their movement and alignment relative to the contact points. This visual representation provides clear evidence of

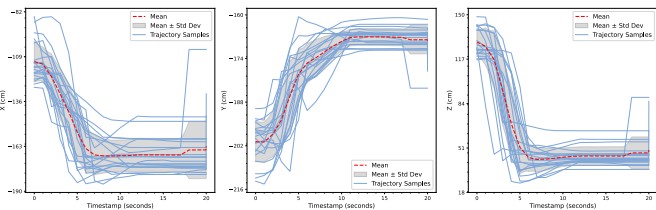

Fig. 9: Randomly sampled grasping trajectories for the middle proximal position.

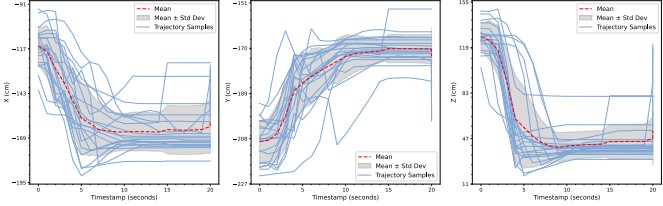

Fig. 10: Randomly sampled pushing trajectories for the middle distal position.

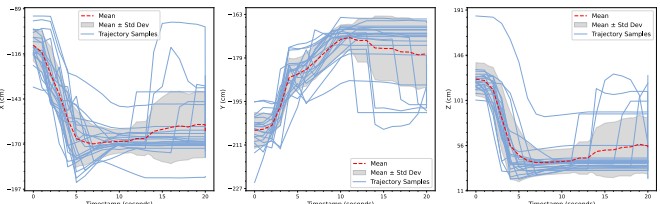

Fig. 11: Randomly sampled pointing trajectories for the index tip position.

the relationship between joint positions and contact points during each type of interaction. The following section will provide a detailed analysis and summary of trajectory similarity, joint orientation, and joint-to-contact position distance.

### A. Trajectory Similarity

Figure 7 illustrates the similarities and differences in joint trajectories using DTW as an evaluation metric for pushing interaction. The indices from 0 to 18 correspond to trajectories recorded from 19 subjects (out of 20), with one subject excluded as an outlier to ensure a fair comparison. Given space constraints, we included only one trajectory for each additional movement direction. Specifically, Index 19 represents the backward trajectory, Index 20 denotes the left-to-right movement, Index 21 corresponds to the right-to-left movement, Index 22 reflects the downward movement, and Index 23 represents the upward movement.

The DTW distance is visualized as a color-coded heatmap, where more intense red shades indicate larger DTW distances, signifying greater dissimilarity between trajectories. Conversely, if two trajectories are identical or highly similar, the DTW distance is zero or close to zero, represented by lighter or less intense colors. As observed in the graph (Fig.

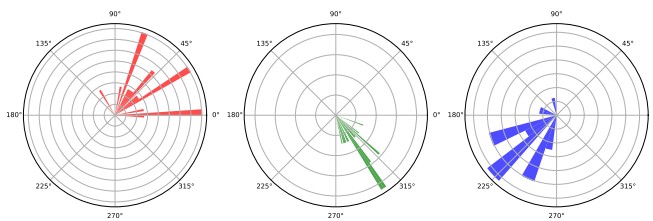

Fig. 12: Roll (left), Pitch (middle), and Yaw (right) orientation during forward grasping.

7), the trajectories corresponding to other movement directions (backward, lateral, and vertical) exhibit significant differences compared to the forward trajectories, as indicated by the strong red coloration. This shows the distinct nature of movement in different directions, showing the collected datasets captured both spatial and temporal differences between trajectories.

Similarly, Fig. 8 shows the joint trajectory similarities for grasping and pointing interactions. Like the pushing interaction, these heatmaps use DTW to measure the differences. The red colors show bigger differences between trajectories, while lighter colors show more similarity. Both grasping and pointing interactions reveal clear differences in movement patterns, depending on the type of interaction and direction.

Additionally, to examine the similarities of joint trajectories across different subjects, we randomly selected data samples for the three interaction types: grasping, pushing, and pointing. Fig. 9 (grasping trajectories), Fig. 10 (pushing trajectories), and Fig. 11 (pointing trajectories) illustrate the randomly sampled trajectories for all subjects. These plots demonstrate that the trajectories follow a consistent pattern across different subjects for each interaction type. This consistency suggests that the dataset captures reliable and repeatable movement patterns, making it a strong resource for analyzing human interactions.

Together, these graphs (Fig. 7, 8, 9, 10, and 11) highlight the ability of the dataset to represent unique and consistent movement patterns for each interaction type, offering valuable insights into both spatial and temporal dynamics. This provides strong evidence that the dataset can be used for analyzing different human interactions with accuracy and detail.

### B. Joint Orientation

The Rayleigh test results for roll, pitch, and yaw during forward grasping shows significant clustering in all three angular orientations, as shown in Fig. 12. The p-values for roll $(6.24 \times 10^{-9})$, pitch $(6.43 \times 10^{-13})$, and yaw $(2.55 \times 10^{-8})$ are all much smaller than 0.05, leading to the rejection of the null hypothesis of uniform distribution. The high Z-statistics for roll (14.903), pitch (18.924), and yaw (14.101) further confirm the presence of strong clustering. This indicates that users exhibit consistent, preferred orientations in roll, pitch, and yaw during forward grasping, with pitch showing the most pronounced alignment. These results highlight the biomechanical

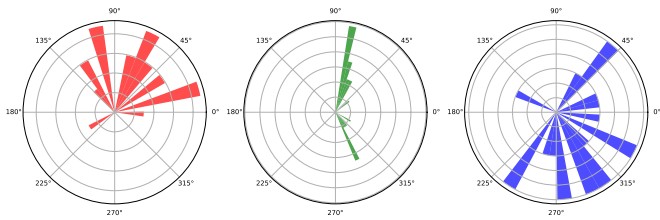

Fig. 13: Roll (left), Pitch (middle), and Yaw (right) orientation during left-to-right grasping.

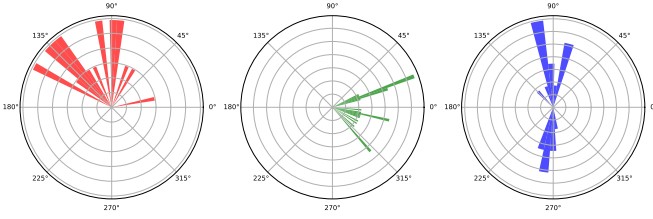

Fig. 14: Roll (left), Pitch (middle), and Yaw (right) orientation during upward grasping.

constraints and natural hand alignment used during grasping actions in MR.

The Rayleigh test results for roll, pitch, and yaw during left-to-right grasping in MR reveal significant clustering in all three angular orientations. The p-values for roll ($1.18 \times 10^{-5}$), pitch (0.0051), and yaw (0.0040) are all smaller than 0.05, leading to the rejection of the null hypothesis of uniform distribution for each angle. The Z-statistics for roll (10.022), pitch (5.056), and yaw (5.268) further confirm the presence of clustering, with roll showing the strongest alignment. As seen in Fig. 13, this indicates that users exhibit consistent preferred orientations in roll, pitch, and yaw during grasping from left to right, with roll displaying the most pronounced alignment. These results highlight the natural alignment of the hand and the movement constraints used during grasping actions in MR.

Similar to other directions, all except for upward grasping shows a preferred direction. In the case of upward grasping, the Rayleigh test indicates significant directional patterns in the Roll and Pitch angles, as both have very small p-values (less than 0.05), suggesting non-uniform distributions and strong directional biases. However, as illustrated in Fig. 14, the Yaw angle's p-value is 0.4056, much larger than 0.05, implying that the Yaw data are uniformly distributed with no significant directional bias. In summary, while the Roll and Pitch angles exhibit clear patterns, the Yaw angle does not show directional preference.

### C. Joint to Contact Position Distance

As seen in Fig. 15 bottom and top left, for pointing upward interaction, 80% of the joints perfectly align with the contact

points (zero distances), while the remaining 20% show small deviations, with two joints having notable distances of 1.80 and 1.07. The mean final joint positions closely match the mean contact point positions along the X-axis (-112.51 vs. -112.14) and Y-axis (-195.21 vs. -195.30), indicating good alignment. In the Z-axis, the mean joint position (188.06) is slightly lower than the mean contact point position (188.23), suggesting a small vertical offset. The standard deviations for joint positions are higher than for contact points across all axes, particularly in the X-axis (15.13 vs. 14.88) and Z-axis (14.02 vs. 13.55), indicating more variability in joint positioning compared to the contact points. Overall, the alignment could best said strong, with only 20% of joints slightly off.

In the middle top and bottom (Fig. 15), the evaluation for pushing forward interaction shows that 55% of the joints perfectly align with the contact points (zero distances), while the remaining 45% show slight deviations. Notably, large deviations are observed for certain interactions, with the highest distances being 8.09, 7.11, 4.58, and 3.29, indicating significant misalignment for these joints. The mean final joint positions are close to the mean contact point positions along the Y-axis (-207.82 vs. -207.13) and Z-axis (166.93 vs. 165.33), but there is a more noticeable offset in the X-axis (-114.08 vs. -110.73), suggesting that joints are positioned slightly further back relative to contact points. The standard deviations are higher for joint positions than contact points across all axes, particularly in the Z-axis (9.69 vs. 8.47), highlighting greater variability in joint positioning. The alignment is generally good, with slight deviations in 45% of the joints.

In the right top and bottom (Fig. 15) for grasping downward interaction, the analysis reveals that 50% of the joints perfectly align with the contact points (zero distances), while the remaining 50% show deviations. Among these, the largest deviations are 3.85, 3.59, and 1.93, indicating areas where alignment could be improved. The mean final joint positions closely match the mean contact point positions along the X-axis (-123.22 vs. -122.78) and Y-axis (-193.93 vs. -193.74), suggesting good horizontal alignment. However, joint positions (126.72) along the Z-axis are slightly higher than the contact points (124.67), indicating a small vertical offset. The standard deviations for joint positions are slightly larger than for contact points on all axes, with the greatest difference in the Z-axis (13.66 vs. 11.62), reflecting greater variability in the joint positions relative to the contact points.

Here, the differences between joint positions and contact points arise from the interaction between the user and virtual objects, which are simplified collisions that do not perfectly match the actual object geometry. Flat collision boundaries in the pushing task make contact detection easier but can cause small shifts with slight hand movements. In grasping, the curved surface of the handle mesh increases variability in contact detection due to changes in hand orientation. These limitations in collision modeling contribute to the observed deviations in joint and contact point alignment.

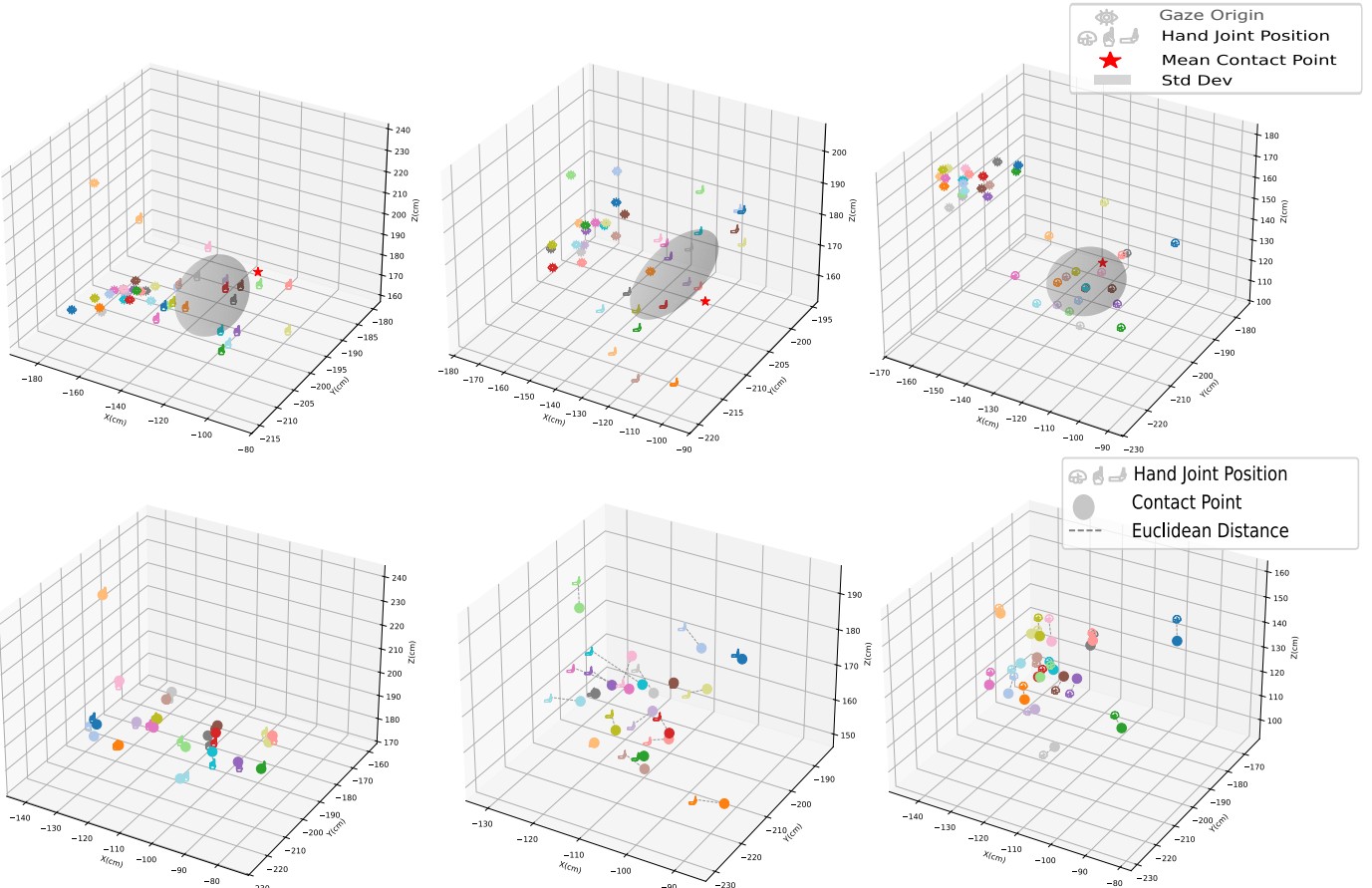

Fig. 15: Euclidean distances between interaction joint positions and contact points, along with the mean and standard deviation for final joint and contact point positions. The top row images correspond to upward pointing (left), forward pushing (middle), and downward grasping (right). Similarly, the bottom row shows pointing (left), pushing (middle), and grasping (right) actions.

## VII. CONCLUSION

This paper presented MiXR-Interact, a dataset of tracked gaze, body, and hand movements in mixed reality (MR), specifically designed for Meta Quest headsets. The dataset contains three core interaction types — pushing, pointing, and grasping — each performed in six different directions to enhance generalizability. By defining 17 precise key points for interaction in each direction, the dataset provides accurate tracking and localization of joint-to-contact positions, offering a resource for researchers and developers in MR and HRI. In addition to the dataset, the paper introduces evaluation metrics for assessing trajectory similarity, joint orientation, and joint-to-contact position differences. The analysis reveals key insights into user movement consistency and joint-to-contact alignment. For trajectory similarity, Dynamic Time Warping (DTW) distances highlight distinct differences in movement direction, with higher DTW values observed in non-forward pushing tasks, indicating greater temporal and spatial variability. The joint orientation analysis shows significant clustering of roll, pitch, and yaw for most interaction types, as confirmed by Rayleigh test results. For joint-to-contact position distance, the evaluation shows that 80% of joints

perfectly align with contact points in upward pointing, while 55% and 50% alignment are observed for forward pushing and downward grasping, respectively. Variability in joint positions, as indicated by higher standard deviations compared to contact points, reflects the influence of collision models and user movement variability. The MiXR-Interact dataset, along with its proposed evaluation metrics and implementation insights, provide a valuable resource for researchers and developers in MR and HRI. By enabling a better understanding and analysis of human movement during interactions, this work supports the development of more precise, adaptive, and responsive MR-based interaction systems.

As future work, we are exploring the development of a deep learning framework to predict and estimate interaction types and points in real time using this dataset. This approach aims to leverage the rich data in MiXR-Interact to train machine-learning models capable of recognizing and forecasting user intent and movement trajectories during MR interactions. Such advancements could enable more adaptive, responsive, and personalized VR and MR experiences.

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
