# OpenReview forum: "MiXR-Interact: Mixed Reality Interaction Dataset for Gaze, Hand, and Body"
_humanrobotinteraction.org/HRI/2025/Workshop/VAM — HRI 2025 Workshop VAM Submission_

### Official Review · Reviewer_9L6S · 2025-02-28

**Rating:** 7
**Confidence:** 5

**Review:**

Summary:
This paper proposes a dataset called MiXR-Interact which is composed of gaze, upper body movements, hand gesture data, and virtual object poses. This combination of data enables machine learning models to learn more complex relationships between users’ body states, ultimately enabling it to predict users’ intentions. Understanding users’ intentions is very important for areas such as teleoperation or learning from demonstration. Data was collected through a user study (N=20) where users would interact with 3D objects: cube for pushing, sphere for pointing, and door handle for grasping. The Trajectory similarity, joint orientation, and joint to contact position distance were then used to evaluate the dataset. This dataset is open source and can be very valuable to the VAM-HRI community.


Strengths:
The dataset is open source, allowing the community to easily access and leverage this work.
Data was collected in a hardware agnostic manner, allowing for more members of the community to be able to leverage this work on their own hardware
User study for collecting data was sound

Areas for Improvement:
This dataset is collected using one object shape for each action, and there isn’t an overlap of shapes across actions (for example, grasping a sphere might be different than grasping a door handle). This paper could benefit from discussing this limitation
Labeling the indices in Figures 7 and 8 would help readers better understand that indices 19-23 are from different movement directions

Small Typos:
The URL text and the hyperlink are mismatched for the first footnote. The hyperlink seems correct, but the URL text is not.

Recommendation:
This work was very interesting, and seems like a great fit to VAM-HRI. In future work it would be interesting to see how well models trained on this data are able to predict users’ intentions. I recommend this paper for acceptance at VAM-HRI this year.

---

### Decision · Program_Chairs · 2025-02-26

Accept